# Effects of Different Salinity Stress on the Transcriptomic Responses of Freshwater Crayfish (*Procambarus clarkii*, Girard, 1852)

**DOI:** 10.3390/biology13070530

**Published:** 2024-07-16

**Authors:** Lei Luo, Li-Shi Yang, Jian-Hua Huang, Shi-Gui Jiang, Fa-Lin Zhou, Yun-Dong Li, Song Jiang, Qi-Bin Yang

**Affiliations:** 1Key Laboratory of South China Sea Fishery Resources Exploitation and Utilization, Ministry of Agriculture and Rural Affairs, South China Sea Fisheries Research Institute, Chinese Academy of Fishery Sciences, Guangzhou 510300, China; leiluo2019@163.com (L.L.); huangjianhua@scsfri.ac.cn (J.-H.H.); jiangsg@21cn.com (S.-G.J.); zhoufalin@aliyun.com (F.-L.Z.); liyd2019@163.com (Y.-D.L.); tojiangsong@163.com (S.J.); 2Shenzhen Base of South China Sea Fisheries Research Institute, Chinese Academy of Fishery Sciences, Shenzhen 518108, China; 3School of Life Sciences, Sun Yat-Sen University, Guangzhou 510275, China; 4Key Laboratory of Efficient Utilization and Processing of Marine Fishery Resources of Hainan Province, Sanya Tropical Fisheries Research Institute, Sanya 572018, China; yangqibin1208@163.com

**Keywords:** salinity, transcriptome, mechanism, osmotic mediation, *Procambarus clarkii*

## Abstract

**Simple Summary:**

*Procambarus clarkii* is an economic freshwater aquaculture species which is popular with consumers for its delicious flavor and high protein content. Salinization of freshwater ecosystems is an increasingly pressing global issue that poses a significant threat to aquaculture. Salinity is an important environmental factor directly affecting the metabolism, growth, reproduction, and physiological processes of aquatic animals. In this study, crayfish were subjected to acute low salt (6 ppt) and high salt (18 ppt) stress and investigated using transcriptome sequencing technology. The response of the crayfish to different salinity stresses, especially immunity, metabolism, ion transport, and osmoregulation, was analyzed to illustrate the resistance mechanism of crayfish facing salt stress. The results of this study are intended to deepen our understanding of the mechanisms by which freshwater organisms respond to salinity stress and provide useful references for the healthy culture of crayfish and the utilization of saline soils.

**Abstract:**

Salinization of freshwater ecosystems is a pressing global issue. Changes in salinity can exert severe pressure on aquatic animals and jeopardize their survival. *Procambarus clarkii* is a valuable freshwater aquaculture species that exhibits some degree of salinity tolerance, making it an excellent research model for freshwater aquaculture species facing salinity stress. In the present study, crayfish were exposed to acute low salt (6 ppt) and high salt (18 ppt) conditions. The organisms were continuously monitored at 6, 24, and 72 h using RNA-Seq to investigate the mechanisms of salt stress resistance. Transcriptome analysis revealed that the crayfish responded to salinity stress with numerous differentially expressed genes, and most of different expression genes was observed in high salinity group for 24h. GO and KEGG enrichment analyses indicated that metabolic pathways were the primary response pathways in crayfish under salinity stress. This suggests that crayfish may use metabolic pathways to compensate for energy loss caused by osmotic stress. Furthermore, gene expression analysis revealed the differential expression of immune and antioxidant-related pathway genes under salinity stress, implying that salinity stress induces immune disorders in crayfish. More genes related to cell proliferation, differentiation, and apoptosis, such as the Foxo, Wnt, Hippo, and Notch signaling pathways, responded to high-salinity stress. This suggests that regulating the cellular replication cycle and accelerating apoptosis may be necessary for crayfish to cope with high-salinity stress. Additionally, we identified 36 solute carrier family (SLC) genes related to ion transport, depicting possible ion exchange mechanisms in crayfish under salinity stress. These findings aimed to establish a foundation for understanding crustacean responses to salinity stress and their osmoregulatory mechanisms.

## 1. Introduction

Freshwater ecosystems provide a wide range of natural resources for fisheries, aquaculture, production, and recreational activities, which are essential for human survival [1]. However, human activities such as urbanization, commercial pollution, and hydrological disturbances severely affect freshwater ecosystems such as rivers, lakes, and wetlands [2,3]. Salinization also exerts pressure on freshwater ecological resources. Global warming has led to rising sea and estuarine levels, and the continued intrusion of saline water into the upper reaches of rivers is one of the main causes of increased salinity in freshwater ecosystems [4,5,6]. Furthermore, human industrial activities, road de-icing salts, and agricultural irrigation have accelerated freshwater resource salinization [7,8]. Groundwater salt rises to the surface via capillary action in arid and semiarid areas, and salts deposited on the soil surface after water evaporation also cause soil salinization [9]. Freshwater ecosystem salinization is a global problem affecting over 100 countries and poses a significant threat to agriculture, negatively impacting biodiversity, species ecosystems, food security, and human economic welfare [10]. Therefore, studying the response of freshwater species to saline stress and the effective exploitation of saline soils are urgent and essential tasks for scientists.

Salinity is an important environmental factor affecting metabolism, growth, reproduction, and physiological processes [11,12,13]. Studies have shown that salinity in aquaculture waters above the species acclimation threshold will lead to histopathological changes in organisms and can also cause immune disorders, osmoregulatory imbalances, disturbances in the gut microbiota, and increased mortality, among other harmful effects [14,15,16]. Aquatic animals respond to the osmotic stress caused by increased salinity through heightened energy metabolism [17]. However, excessive energy metabolism alters oxidative levels, producing reactive oxygen species (ROS) accumulation, possibly triggering oxidative stress injury [18]. Furthermore, salinity stress affects the ion transport processes and transport-related enzyme activities, leading to disturbances in the osmotic regulation of salt ions in organisms [19]. Among all aquatic animals, crustaceans are the most sensitive to salinity fluctuations due to their osmoregulatory system’s inefficiency, especially in narrowly saline species [20]. Although the effects of salinity fluctuations on crustaceans have become a popular topic, most of them are only detected using macroscopic indicators (e.g., enzyme activities and changes in gene expression) [21,22,23], and the microscopic internal regulatory mechanisms are still unclear. Therefore, elucidating these regulatory mechanisms will help us understand whether crustaceans can cope with further salinity fluctuations, is important for screening salt-tolerant species of aquatic animals, and provides a scientific basis for exploring the aquaculture of aquatic species in saline soils.

High-throughput RNA sequencing technology has been widely used in various organisms and has been successfully used to analyze many aquatic species in response to salinity stress. Jiang et al. established a transcriptome database of pufferfish under long-term low-salinity stress and screened three solute carrier family (SLC) genes related to ion transport [24]. Liu et al. analyzed the liver transcript levels of halibut under low-salinity stress and showed that salinity stress affects the liver lipid metabolism of halibut [25]. Through transcriptome sequencing, Chen et al. identified various genes involved in osmoregulation under different salinity stresses in *Oratosquilla oratoria* [26]. Although considerable progress has been made in the studying of transcript levels in salinity-stressed aquatic animals, relatively few studies have been conducted on freshwater, narrowly saline crustaceans, especially at the transcript level.

*Procambarus clarkii* is a freshwater crayfish native to the United States and Mexico and was introduced to China from Japan in the 1920s [27]. Its delicious meat and high protein content has made it widely welcomed by consumers. Crayfish production was ~2.89 million tons in 2022 in China, making it the largest crustacean aquaculture species in China [28]. Therefore, in the present study, crayfish were subjected to salinity stress and investigated using transcriptome sequencing technology. The response of the crayfish to different salinity stresses, especially immunity, metabolism, ion transport, and osmoregulation, was analyzed to illustrate the mechanism of crayfish resistance to salinity. The results of this study are intended to deepen our understanding of the mechanisms by which freshwater organisms respond to salinity stress and provide useful references for the healthy culture of crayfish and the utilization of saline soils.

## 2. Materials and Methods

### 2.1. Experimental Animal

Healthy *P. clarkii*, with a body length of 6.0 ± 0.5 cm and a weight of 15.0 ± 2.0 g, were purchased from Jian Li, China. They were temporarily reared in the recirculating water system of the Shenzhen Test Base of the South China Sea Fisheries Research Institute, Chinese Academy of Fishery Sciences (Shenzhen, China), and fed 3% of their body weight with commercial feed (China Hubei Tianbang Feed, Tianmen, China). The water temperature was maintained at 25 ± 1 °C, with a pH of 7.5 and dissolved oxygen levels of 6.5 ± 0.5 mg/L.

### 2.2. Exposure and Sample Collection

According to the safe salinity (6 ppt) of *P. clarkii* (body weight about 15 g) and our previous results of the 96 h semi-lethal salinity experiment (21 ppt) [29,30], the crayfish were marked with different labels as follows: control group (NC, 0 ppt), low-salt group (LS, 6 ppt), and high-salt group (HS, 18 ppt). A total of 270 healthy crayfish were selected and divided into three groups of 90 crayfish each. The experimental salt water was prepared in proportions of seawater and fresh water, and the water was not replaced during the test. The hepatopancreas tissues of three crayfish of each group were dissociated from the carapace at 6 h, 24 h, 72 h, respectively, and frozen in liquid nitrogen for transcriptome sequencing.

### 2.3. RNA Extraction, Library Preparation, and Sequencing

The Trizol reagent kit (Invitrogen, Carlsbad, CA, USA) was used to extract total RNA following the manufacturer’s protocol. RNA quality was assessed using an Agilent 2100 Bioanalyzer (Agilent Technologies, Palo Alto, CA, USA) and RNase-free agarose gel electrophoresis. The NEBNext^®^ Ultra^TM^ RNA Library Prep Kit (Ipswich, MA, USA) was used to construct mRNA-seq libraries for all the samples, which were then sequenced on an Illumina HiSeqTM 4000 by Gene Denovo Biotechnology Co. (Guangzhou, China). High-quality reads were produced by fastp software (version 0.18.0) to remove reads that contained adapters [31], poly nucleotides (N), and low-quality bases. Thus, contigs and unigenes were obtained by a short reads assembling program—Trinity software (version: r20140413p1) [32].

### 2.4. Data Analysis

#### 2.4.1. Gene Differential Expression and Functional Annotation

The unigene expression was calculated and normalized using RPKM (reads per kb per million reads) [33]. Differential expression analysis of RNAs was performed between the two groups using the DESeq2_1.20.0 software (and between the two samples using edgeR) [34,35]. Genes were considered differentially expressed if they had a false discovery rate (FDR) parameter of less than 0.05 and an absolute fold change of ≥2. All unigenes were used to analyze gene function with Blastx alignment based on the following databases: NCBI non-redundant protein (Nr) database, Swiss-Prot protein database, Kyoto Encyclopedia of Genes, Genomes (KEGG) database, COG/KOG database, and Gene Ontology (GO) database.

#### 2.4.2. GO, KEGG Enrichment, and Gene Expression Pattern Analysis

To explore how *P. clarkii* responds to salinity fluctuations, we performed GO classification and KEGG pathway annotation of all the differentially expressed genes obtained. Firstly, we used the Blast2GO program (v2.3.5) to identify the characteristics and biochemical metabolic pathways of the DEGs and their products, and the threshold parameter for significant differences was Q ≤ 0.05. Subsequently, we mapped the unigenes to the KEGG database to find the pathways to which they belonged and further determined the unigenes of the ion transport-related pathway.

The expression levels were then normalized to different values to illustrate the expression pattern of the DEGs. In brief, the expression level of each sample was normalized to 0, log2 (v1/v0), log2 (v2/v0), and the values were clustered by Short Time-series Expression Miner software 1.3.8 (STEM 1.3.8). The cluster profile was constructed by three rules as follows: At first, the maximum unit change between the different time points would be 1. Then, the maximum output profiles should be set to 20 (similar expression pattern profiles should be merged). At last, the minimum ratio of the fold change in the DEGs should be more than 2.0. The profiles with *p*-value ≤ 0.05 were considered as significant differences.

#### 2.4.3. Ion Transport-Related Gene Selection and Function Prediction

To understand the differential expression of ion transport-related genes at seven salinity time points, based on the detailed information of the Nr-annotated DEGs, we selected unigenes whose annotation information was functionally annotated as “ion channel” and “transporter”. Next, the functional characteristics of all ion transport genes were described using the GeneCard database (http://www.genecards.org/, accessed on 22 August 2023).

### 2.5. Verification by Quantitative Real-Time PCR (qRT-PCR)

To ensure the reliability of the RNA sequencing results, we randomly selected 9 DEGs for verification by quantitative RT-PCR at two different salinity time points. Using 18S rRNA as a reference gene, Primer Premier 6.0 software was used to design the gene-specific primer pairs (Appendix A). The experiments were performed on a Roche Light Cycler 480 thermal cycler (Roche Applied Science, Penzberg, Germany). The total volume of the reaction solution was 12.5 μL containing 1 μL cDNA, 0.5 μL upstream and downstream primers, 6.25 μL 2×SYBR Premix ExTaq, and 4.25 μL sterile water. qRT-PCR was performed as follows: Five qPCR replicates for each gene, with each replicate using independently prepared RNA samples, and each sample from three technical replicates. The experimental procedure was 95 °C for 2 min. Running conditions were 95 °C for 2 min, 1 cycle, then 95 °C for 15 s, and 60 °C for 1 min, 40 cycles. Amplicons were verified by melting curve analysis and relative expression was determined using the comparative CT method (2^−ΔΔCT^) [36].

### 2.6. Statistical Analysis

All biological experiments were repeated three times independently, and the data analyzed were expressed as mean ± standard deviation (SD). Significant differences were analyzed using Duncan’s multiple comparison test followed by GraphPad Prism software 9. *p* < 0.05 was defined as statistically significant.

## 3. Results

### 3.1. RNA-Seq Data

A total of 884,590,210 raw reads were obtained from seven groups of crayfish hepatopancreas treated at different salinities and time points (Table 1). After data filtering, we obtained 882,552,854 clean reads, accounting for an average of 99.76% of the raw reads. The average Q20% and Q30% were 96.62% and 91.18%, respectively, and the average GC content was 51.84% (Table 1). Clean reads were assembled using Trinity software, and 52,533 unigenes were generated with an average length of 1000 bp, the maximum length of 32,384 bp, and a minimum length of 201 bp. The number and length of N50 were 8596 and 1688 bp, respectively, with an average GC content of 43.18% (Table 2). Based on the protein databases, the gene annotation results showed that 16,957 unigenes were annotated. Of these, 16,893, 8908, 12,210, 8234, and 4816 unigenes showed significant matches with sequences in the Nr, Swiss-Prot, KEGG, COG/KOG, and GO databases, respectively (Appendix A).

### 3.2. Differential Expression Responses of P. clarkii Exposed to Different Salinity Gradients

To investigate the genes that respond to changes in salinity, we compared DEG counts between the six salinity treatment groups and the control group. Compared with the control group, 292 (184 up and 108 down), 622 (280 up and 342 down), 614 (425 up and 189 down), 324 (214 up and 110 down), 2545 (1368 up and 1177 down), and 613 (299 up and 314 down) unigenes were obtained at LS-6h, LS-24, LS-72, HS-6h, HS-24h, and HS-72h, respectively (Figure 1). We then performed GO term enrichment analysis on the DEGs from the six treatment groups to explore how salinity fluctuations affected physiological processes in crayfish (Figure 2). The results indicated that differentially expressed genes were mainly enriched in biological processes (mainly single organisms and metabolic processes), cellular components (mainly cells and cell parts), and molecular functions (mainly catalytic activity and binding).

KEGG enrichment analysis revealed the top 20 pathways in which the DEGs were enriched (Figure 3). Of the three low-salt treatment groups, DEGs were mainly enriched in pyruvate metabolism (Q = 0.010) and glycolysis/gluconeogenesis (Q = 0.035) pathways within the LS-6h group. The DEGs in the LS-24h group were primarily enriched in the pentose and glucuronate interconversion (Q = 0.000), ascorbate and aldarate metabolism (Q = 0.000), drug metabolism—cytochrome P450 (Q = 0.000), metabolism of xenobiotics by cytochrome P450 (Q = 0.000), and retinol metabolism (Q = 0.000) pathways. However, fewer DEGs were significantly enriched in the LS-72h treatment group, mainly in drug metabolism—other enzymes (Q = 0.033). In the high-salinity treatment groups, the DEGs were significantly enriched in 19 pathways at HS-24h, including the pentose and glucuronate interconversions (Q = 0.000) and metabolic pathways (Q = 0.000). Furthermore, DEGs in the ECM–receptor interaction (Q = 0.002) and ascorbate and aldarate metabolism (Q = 0.028) pathways were significantly enriched in the HS-6h and HS-72h treatment groups.

### 3.3. Trend Analysis of DEGs under Salinity Treatment

To visually observe the expression patterns of transcripts in relation to salinity, we analyzed the expression trends of genes in the different salinity treatment groups and performed hierarchical clustering of genes enriched in different pathways (Figure 4 and Figure 5).

Based on the normalization of gene expression trends, 20 expression patterns of 1231 genes were identified under low salt stress (Figure 4A). Among them, seven categories of gene expression patterns showed an overall post-change upregulation trend with increasing salinity stress duration, including profiles of 17 (316 genes), 15 (107 genes), 18 (61 genes), 19 (44 genes), 12 (31 genes), 10 (15 genes), and 6 (10 genes). Similarly, the following seven categories of gene expression patterns showed an overall post-change downregulation trend: profile 1 (165 genes), profile 7 (79 genes), profile 2 (61 genes), profile 13 (28 genes), profile 9 (23 genes), profile 4 (8 genes), and profile 0 (7 genes). In addition, six categories of genes, profile 16 (101 genes), profile 8 (44 genes), profile 3 (36 genes), profile 5 (27 genes), profile 14 (22 genes), and profile 11 (12 genes), showed an overall unchanged expression trend. Furthermore, 36 genes that were significantly differentially expressed in different pathways and signaling pathways were identified using cluster analysis. These included metabolic pathways (glucose, lipid, protein, vitamin, ammonia, and nitrogen metabolism) (Figure 4B), immune and antioxidant pathways, and ABC transporter signaling pathways (Figure 4C).

Similarly, 3133 genes with 20 different expression trends were characterized under high salt stress (Figure 5A). Of these, seven categories of gene expression patterns showed a general up-regulation trend, including profiles 18 (579 genes), 17 (212 genes), 12 (64 genes), 19 (33 genes), 15 (33 genes), 10 (19 genes), and 6 (7 genes). Profiles 1 (525 genes), 7 (248 genes), 2 (193 genes), 0 (151 genes), 9 (62 genes), 4 (57 genes), and 13 (6 genes) showed an overall post-change downregulation trend. Moreover, six gene categories, Profile 11 (379 genes), Profile 16 (265 genes), Profile 8 (173 genes), Profile 14 (58 genes), Profile 3 (40 genes), and Profile 5 (29 genes) showed overall unchanged expression trends. A total of 96 significant differentially expressed genes were clustered into 14 pathway classes. In addition to the metabolic, immune, and antioxidant pathways (Figure 5B), several important signaling pathways were included (Foxo, Hippo, MAPK, Notch, Wnt Toll, and Imd signaling pathways, phosphatidylinositol signaling system, and longevity regulating pathway) (Figure 5C).

### 3.4. Analysis of Genes Related to Ion Exchange in P. clarkii

Based on the screening of transcript data and gene annotation, we identified 36 solute carrier family (SLC) genes related to ion transport that were differentially expressed in the six salinity treatment groups (Table 3 and Figure 6). These genes include SLC41A1 (Magnesium ion transporter), SLC40A1 (Iron ion transporter), SLC39A11/9/2 (Zinc transporter), SLC30A9/7/6/5/2/1 (Zinc transporter), SLC26A11/6 (chloride–bicarbonate exchanger), SLC22A8/6/5/3/15B (Organic cation transporter), SLC13A5 (Na^+^-dependent carboxylate and sulfate transporter), SLC12A9/6/2 (Cation-coupled Cl-Cotransporter), SLC10A/6/3/2 (Ileal sodium/bile acid cotransporter), SLC9A9/8/3R1/2 (Sodium/hydrogen exchanger), SLC8A2/1 (Sodium–calcium exchanger), SLC5A3/1 (Sodium-dependent vitamin transporter), and SLC4A10/1 (Sodium bicarbonate cotransporter).

### 3.5. Quantitative Real-Time PCR

To confirm the RNA-Seq results, nine DEGs were randomly selected for RT-qPCR validation in two salinity groups (Figure 7). Among these, five genes were upregulated (*SLC5A1*, *SOD-4*, *DUOX*, *CHER*, and *TPI1B*) (Figure 7A,C,E,F,H), and four genes were downregulated (*AO2*, *HSP70*, *RGN*, and *SORD*) (Figure 7B,D,G,I). The gene expression trend was similar to that of the transcriptome sequencing results, indicating that the sequencing results were reliable.

## 4. Discussion

Salinity is an important environmental factor, and changes in salinity affect aquatic animal growth and development, metabolism, immunity, and numerous physiological processes [11,12]. In their natural habitats, crustaceans are constantly exposed to a wide range of salinities, making their survival extremely difficult because they lack effective osmoregulators [37]. Understanding the mechanisms by which crustaceans respond to environmental salinity fluctuations is an increasingly important scientific issue. Therefore, in the present study, a freshwater economic crayfish species was used as a research model, and its response pathways to different salinities and ion osmoregulation mechanisms were investigated by transcriptome sequencing.

In this study, we set a low salt concentration of 6 ppt and a high salt concentration of 18 ppt stress on crayfish. The results showed that the differentially expressed genes showed a tendency to increase and then decrease with the prolongation of time under both high and low salt stress, with the most remarkable response to the low salt stress of 24 h. Although crayfish temporarily adapted to these stresses, they still posed significant stress challenges, which explains why crayfish only survive for a short period in many lake inlets [38]. To further determine which pathways were enriched for functional genes under salinity stress, we performed GO functional enrichment analyses on six comparator groups. These showed that the upregulated and downregulated genes were significantly enriched in metabolic processes, cellular components, catalytic activity, and binding processes. Crustaceans respond to environmental stress by compensating for energy loss through metabolic pathways [39,40,41]. A study of the response of mud crabs to salinity fluctuations also showed that enzyme catalysis regulates ionic changes and osmotic pressure [42]. Numerous ion transporters bind and transport salt ions to alleviate the difference in salt ion concentration inside and outside the cell membrane [19]. Similarly, KEGG enrichment analyses showed that metabolic pathways responded most strongly to salinity stress, which was consistent with the results of the GO analyses, revealing that metabolic processes are important in the response to salinity fluctuations in crayfish. Notably, differentially expressed genes were also significantly enriched in ECM–receptor interactions under high salinity stress (Q = 0.002). Therefore, we hypothesized that high salt levels induce a cellular stress response that activates functional genes to regulate cellular responses and ultimately prevent cell death [43,44].

Under varying salinity pressures, crayfish require the synergistic expression of numerous genes to respond to the hypertonic external environment. In our study, 1132 genes were categorized into 20 different expression patterns under low salt (6 ppt) stress, while there were 3133 genes under high salt (18 ppt) stress. This demonstrates that high salt levels induce a more dramatic response and that crayfish require more genes to be differentially expressed to cope with osmotic stress [15]. Gene clustering analyses indicated that metabolic pathways were important in the response to salinity stress in crayfish. This was consistent with our GO and KEGG enrichment analyses, suggesting that metabolic pathways may be necessary for crayfish to compensate for energy loss due to hypertonic environments. Furthermore, changes in immune- and antioxidant-related genes were also found in the salinity results, possibly suggesting that salinity stress leads to immune disorders in crayfish; similar findings have been reported in various aquatic animals [15,45,46]. MAPKs are key signal transducers from the cell’s surface to the inside of the cell and are activated to control many key processes in cell physiology, including cell growth, differentiation, and adaptation to environmental pressures [47]. Our results showed that the differential expression of genes related to the MAPK signaling pathway suggests that the MAPK signaling pathway may be essential for organisms to cope with environmental stress. Two ATP-binding protein genes were differentially expressed under low-salinity stress, implying that crayfish must rapidly translocate ATP to counteract hyperosmotic stress [48]. Notably, crayfish require additional signaling pathways to respond to high-salinity stress. The FoxO signaling pathway is an important biological regulatory network, and the activation of the FoxO signaling pathway can have important roles in metabolic regulation, disease genesis, and cell proliferation [49]. We discovered that FoxO signaling pathway genes were differentially expressed at different time points during high salinity stress, suggesting that high salinity stress may lead to reduced metabolism and increased fatty acid oxidation and glycolysis in crayfish to counteract metabolic diseases; a similar report has been published on *Oreochromis mossambicus* [50]. The Hippo, Notch, and phosphatidylinositol signaling pathways play important roles in the cell differentiation, proliferation, and apoptosis pathways [51,52,53]. The related pathway genes in our results were also differentially expressed at different time points under high salinity stress, suggesting that high salinity stress influences the survival of cells, and that the organism adjusts the cell cycle and promotes cell apoptosis to adapt to the hyperosmotic environment outside the cell membrane. Additionally, the Wnt signaling pathway and Toll and IMD signaling pathway genes were also differentially expressed, which are related to intercellular communication and pathogen recognition, and are similarly expressed in many aquatic animals during increased environmental stress [54,55,56]. Notably, longevity regulatory pathway genes were differentially expressed under high salt stress, which is rare in crustaceans. The longevity regulatory pathway is a complex regulatory network involving numerous signaling pathways, such as the mitochondrial, insulin, and reproductive pathways [57]. It has been reported in higher animals [58] but is less frequently reported in aquatic animals in response to environmental stress. Further, our results revealed that most genes under high salt stress had a significant degree of differential change (sharply upregulated or downregulated) at HS-24h and a tendency to regress at HS-72h. This suggests that high salinity treatment of the crayfish for a period of time (at 24 h) resulted in peak stress, and that the crayfish required more rapid changes in gene expression to regulate body functions in response to the hypertonic environment and produced transient acclimatization and survival, followed by gene expression callbacks. However, failure to acclimatize to high salinity may have led to mortality in the crayfish [59]. Similar findings were reported for the transcript of *Oreochromis niloticus* in response to salinity stress [60].

In aquatic animals, osmotic homeostasis under salinity stress is regulated by multiple inorganic ion exchanges both inside and outside the cell membrane [61,62]. This study revealed the differential expression of genes related to ion exchange in crayfish under low and high salinity gradients in seven groups. To visualize the ion regulatory roles of these genes, we simplified all the genes related to ion exchange into a single diagram (Figure 8). Sodium (Na^+^) and chloride ions (Cl^−^) account for more constituents of the hemolymph of crustaceans [11,42,62]. Thus, when crayfish are exposed to salinity stress, osmotic pressure is balanced by the preferential uptake or secretion of Na^+^ and Cl^−^. Our results revealed that SLC12A2/6/9, which are mainly responsible for the transport of sodium chloride ions [63], were differentially expressed under salinity stress; in particular, SLC12A2 was significantly upregulated at HS-24h, suggesting that crayfish under acute salinity stress may first reduce salt loss through the active uptake of extramembrane sodium chloride and potassium ions. Notably, the upregulated expression of SLC9A9/8/2/3R1 may generate a steep electric potential difference by enhancing H^+^ release (hydration), which, in turn, promotes Na^+^ uptake [64]. Under low-salinity conditions, we found that the expression of SLC8A2/1 increased. New evidence suggests that SLC8A2/1 is involved in the removal (Ca^2+^) from cells, followed by the release of only 2Na^+^ [65]. We also found that SLC4A10/1 and SLC26A11/6 were initially upregulated and subsequently downregulated under salinity stress. Earlier studies have suggested that SLC4A10/1 may increase the co-transport of Na^+^ and bicarbonate (HCO_3_^−^), allowing more Na^+^ to be transported into the plasma [66]. Simultaneously, SLC26A11/6 can promote Cl^−^ uptake and HCO_3_^−^ secretion [67]. Specifically, a high positive transepithelial potential would be generated in the plasma owing to the rapid influx of large amounts of Na^+^ [68]. To drive Na^+^ out of the plasma, SLC4A10/1 was downregulated at HS-72h to slow this trend [62,66]. Additionally, seven sodium-dependent biomolecule co-transporter genes (SLC13/10/5 family) were also identified, among which SLC5A1 (sodium-dependent glucose transporter) expression was significantly elevated under salinity stress, which once again corroborated that rapid transmembrane transport and constant catabolism of glucose to provide ATP quickly are essential for crayfish to cope with osmotic stress [68]. Finally, several genes involved in the transport of Fe^2+^, Zn^2+^, Mg^2+^, and other cations were identified. Among them, Mg^2+^, Fe^2+^, and Zn^2+^ may activate other ion channels by altering catalytic processes or cellular structures [69,70]. Previous studies have suggested that some inorganic cations may play a role in the regulation of cell morphology during osmotic control, indirectly influencing ion secretion and uptake [71]. Overall, we hypothesized that changes in the salt stress levels dominate the ion transport patterns of *P. clarkii.* The synergistic action of various ion transport channels triggers a complex regulatory strategy that ultimately improves tolerance to salt stress.

## 5. Conclusions

Freshwater salinization is a pressing global environmental problem. Excessive salinity severely affects the metabolism, osmoregulation, behavior, and reproduction of organisms, ultimately leading to a serious loss of biodiversity in freshwater ecosystems. In this study, RNA-Seq analyses were performed on *P. clarkii* under acute high and low salt stress, and more differentially expressed genes were identified under high-salt stress than under low salt stress. This evidence suggests that high salt concentrations may induce more intense osmotic stress than low salt concentrations. Our results also suggest that salinity stress alters the rates of glucose, lipid, and amino acid metabolism and may ultimately enhance osmoregulation in crayfish by regulating the rates of energy metabolism and ion transmembrane transport. Moreover, immune- and antioxidant-related pathway genes were differentially expressed under salinity stress, suggesting that salinity stress induces immune disorders in crayfish. Importantly, cell proliferation, differentiation, apoptosis, and related signaling pathways (Foxo, Wnt, Hippo, Notch signaling pathway, etc.) responded to high salinity stress in crayfish compared to low-salinity stress, revealing that regulating the cell replication cycle and accelerating apoptosis may be necessary for crayfish to respond to high salinity stress. We identified 36 solute carrier family (SLC) genes related to ion transport, indicating possible ion exchange mechanisms in crayfish exposed to salinity stress. The results of the study elaborate on the response mechanism of crayfish to salinity, provide a molecular theoretical basis for the domestication of crayfish for low-salt adaptation, and provide data support for the development and utilization of saline soils.

## Figures and Tables

**Figure 1 biology-13-00530-f001:**
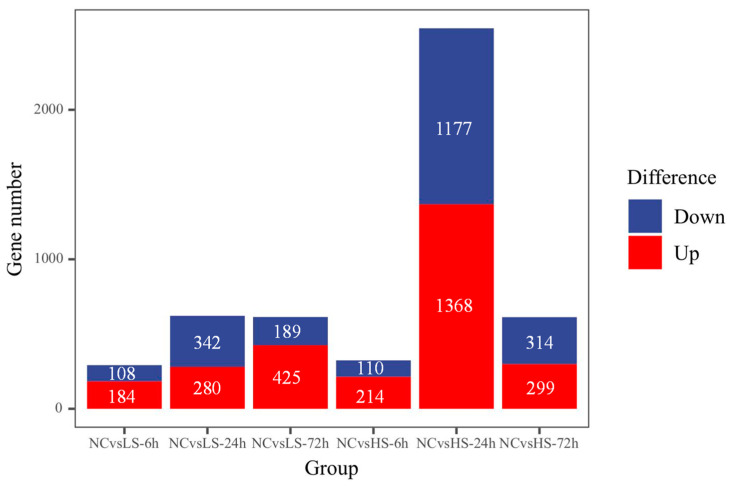
Statistics of DEGs in different salinity treatment groups.

**Figure 2 biology-13-00530-f002:**
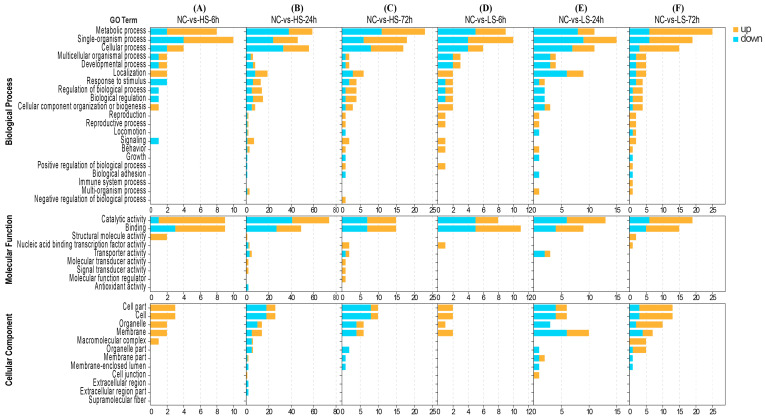
GO functional annotations of DEGs for *P. clarkii* exposed to salinity fluctuation; (**A**): NCvsHS-6h, (**B**): NCvsHS-24h, (**C**): NCvsHS-72h; (**D**): NCvsLS-6h, (**E**): NCvsLS-24h, (**F**): NCvsLS-72h.

**Figure 3 biology-13-00530-f003:**
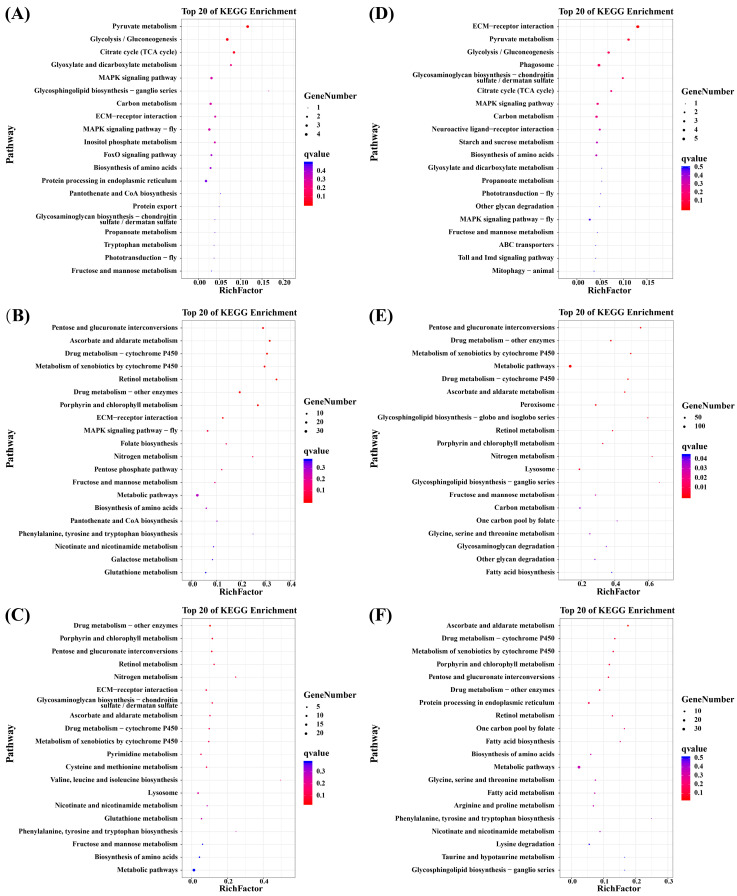
KEGG enrichment analysis results of DEGs of *P. clarkii* under salt stress; (**A**): NCvsLS-6h, (**B**): NCvsLS-24h, (**C**): NCvsLS-72h; (**D**): NCvsHS-6h, (**E**): NCvsHS-24h, (**F**): NCvsHS-72h.

**Figure 4 biology-13-00530-f004:**
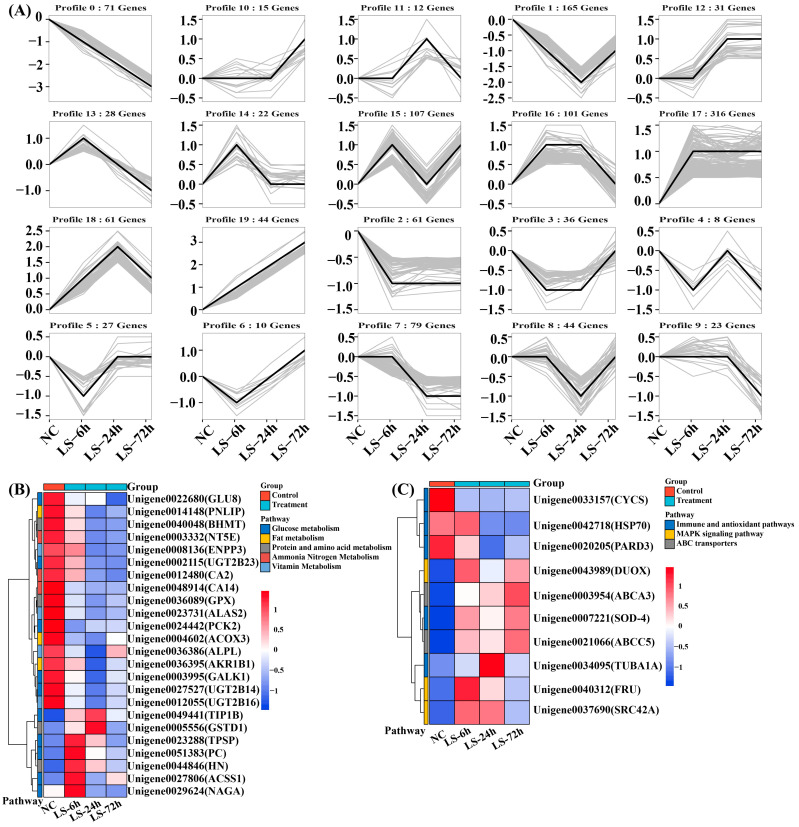
Analysis of gene expression patterns under low salt stress based on normalization of gene expression trends. (**A**): gene expression trend analysis; (**B**): gene expression clustering pattern analysis of metabolic pathways; (**C**): gene expression clustering pattern analysis of other signaling pathways. (The color scale on the right ranges from the lowest (blue) to the highest (red) expression level). The detailed gene information in the right range is shown in Appendix A.

**Figure 5 biology-13-00530-f005:**
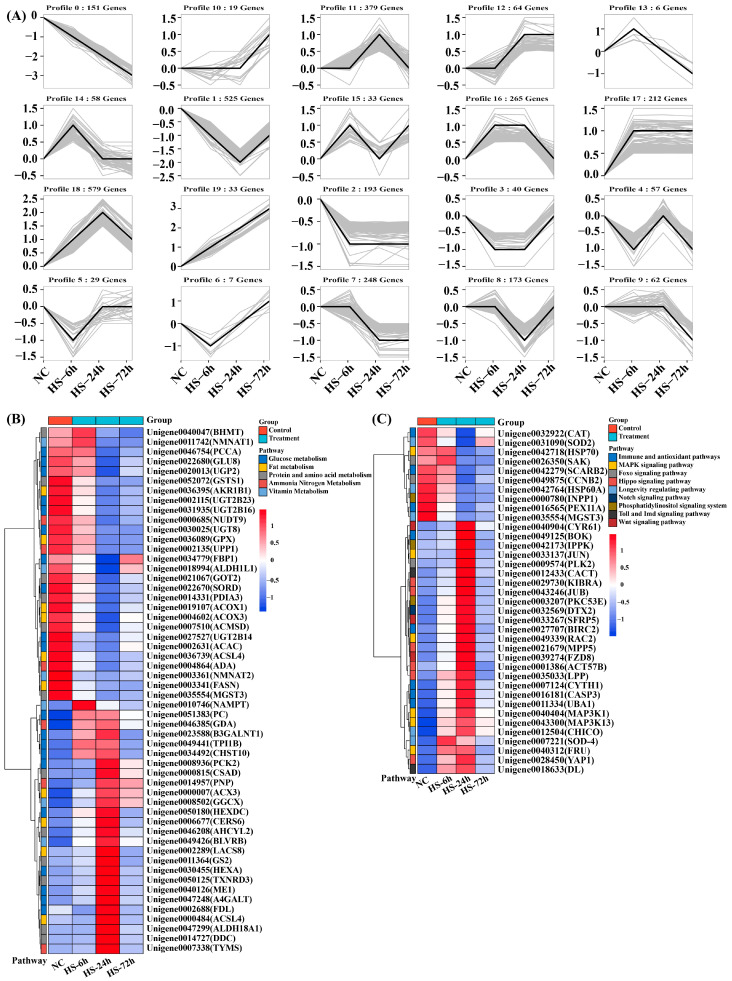
Analysis of gene expression patterns under high salt stress based on normalization of gene expression trends. (**A**): gene expression trend analysis; (**B**): gene expression clustering pattern analysis of metabolic pathways; (**C**): gene expression clustering pattern analysis of other signaling pathways. (The color scale on the right ranges from the lowest (blue) to the highest (red) expression level). The detailed gene information in the right range is shown in Appendix A.

**Figure 6 biology-13-00530-f006:**
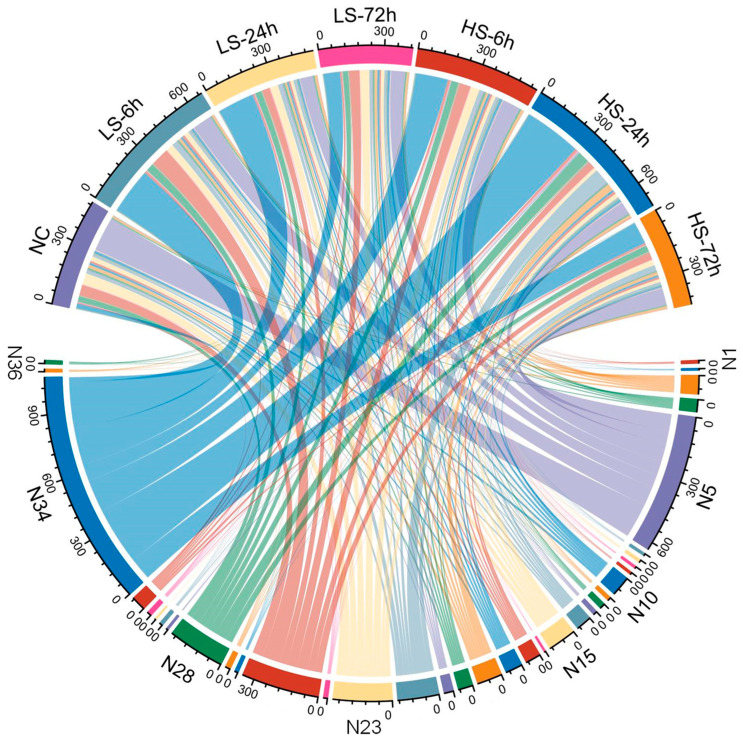
Chordal graph of 36 solute carrier family genes under different salinity treatments. N1–N36 represent the genes of Number 1–36 in Table 3, the genes correspond to the treatment groups one by one, the wider the chord width, the higher the gene expression.

**Figure 7 biology-13-00530-f007:**
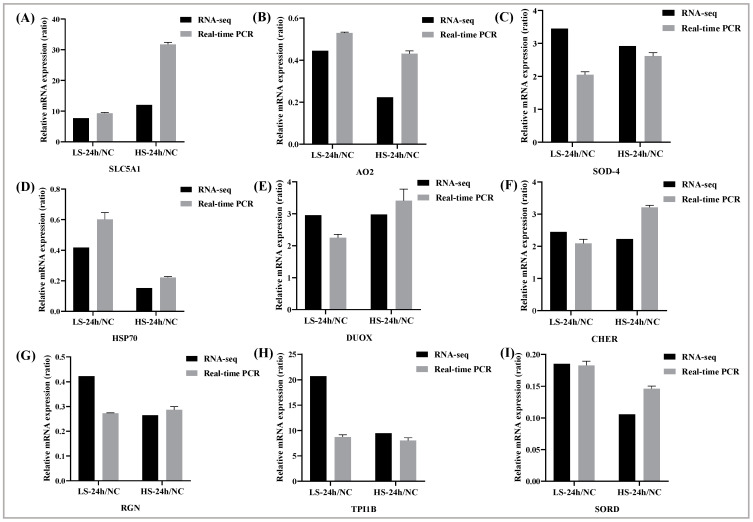
Validation of DEGs with qRT-PCR. Relative mRNA expression level < 1 indicates downregulation, and relative mRNA expression level > 1 indicates upregulation. (**A**): *SLC5A1*, (**B**): *AO2*, (**C**): *SOD-4*, (**D**): *HSP70*, (**E**): *DUOX*, (**F**): *CHER*, (**G**): *RGN*, (**H**): *TPI1B*, (**I**): *SORD*. The data of this study are presented as mean ± SD of three parallel measurements.

**Figure 8 biology-13-00530-f008:**
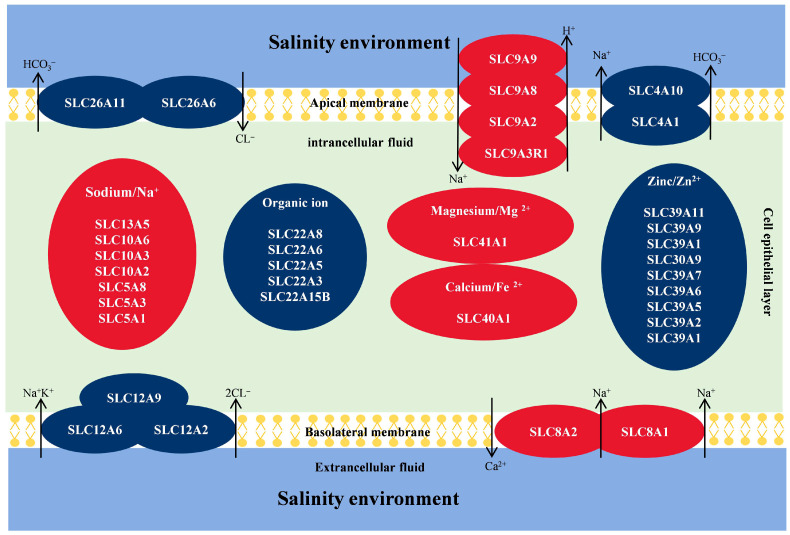
Schematic of ion exchange-related genes in *P. clarkii* exposed to salinity environment.

**Table 1 biology-13-00530-t001:** Sequencing sample data and quality control.

Sample	Raw Data	Clean Data (%)	Q20 (%)	Q30 (%)	GC (%)
NC-1	43,079,546	42,982,162 (99.77%)	96.51%	90.75%	51.68%
NC-2	38,593,886	38,496,476 (99.75%)	96.17%	90.06%	51.31%
NC-3	43,843,540	43,725,484 (99.73%)	96.80%	91.58%	51.91%
LS-6h-1	44,961,252	44,810,234 (99.66%)	96.41%	90.82%	51.87%
LS-6h-2	40,666,280	40,580,188 (99.79%)	96.61%	91.15%	52.15%
LS-6h-3	45,471,462	45,335,054 (99.70%)	96.56%	91.11%	50.69%
LS-24h-1	36,343,018	36,273,260 (99.81%)	96.75%	91.39%	52.11%
LS-24h-2	37,333,288	37,262,908 (99.81%)	96.61%	91.05%	51.67%
LS-24h-3	43,514,756	43,385,520 (99.70%)	96.48%	91.04%	50.90%
LS-72h-1	41,937,250	41,860,300 (99.82%)	96.72%	91.26%	50.97%
LS-72h-2	39,298,830	39,218,856 (99.80%)	96.96%	91.88%	52.03%
LS-72h-3	39,176,456	39,097,798 (99.80%)	96.30%	90.59%	52.02%
HS-6h-1	45,496,514	45,380,462 (99.74%)	96.26%	90.52%	50.76%
HS-6h-2	37,971,054	37,894,032 (99.80%)	96.69%	91.36%	53.46%
HS-6h-3	44,680,448	44,583,988 (99.78%)	96.63%	90.88%	51.51%
HS-24h-1	40,967,824	40,872,802 (99.77%)	96.24%	90.51%	52.03%
HS-24h-2	41,390,576	41,287,814 (99.75%)	96.22%	90.49%	52.74%
HS-24h-3	46,509,192	46,401,214 (99.77%)	98.00%	94.48%	52.18%
HS-72h-1	41,216,070	41,140,894 (99.82%)	96.42%	90.79%	52.28%
HS-72h-2	48,341,970	48,254,082 (99.82%)	96.89%	91.75%	52.41%
HS-72h-3	43,796,998	43,709,326 (99.80%)	96.70%	91.35%	52.01%

**Table 2 biology-13-00530-t002:** Quality statistics of unigenes assembly.

Genes Number	GC (%)	N50 Number	N50 Length	Max Length	Min Length	Average Length	Total Bases
52,533	43.18	8596	1688	32,394	201	1000	52,553,090

**Table 3 biology-13-00530-t003:** The 36 SLC family (solute carrier family) genes related to ion transport were differentially expressed in six salinity groups.

Number	Symbol	Description	Rpkm (Mean)						
			NC	LS- 6h	LS- 24h	LS- 72h	HS- 6h	HS- 24h	HS- 72h
1	SLC41A1	Solute carrier family 41 member 1 (Magnesium ion transporter)	1.21	1.11	1.96	0.96	1.12	3.69	0.83
2	SLC40A1	Solute carrier family 40 member 1 (Iron ion transporter)	1.50	0.76	0.39	0.86	0.89	0.61	0.68
3	SLC39A9	Solute carrier family 39 member 9 (Zinc transporter ZIP9)	9.99	11.04	11.65	11.11	12.65	12.88	13.12
4	SLC39A11	Solute carrier family 39 member 11 (Zinc transporter ZIP11)	6.87	7.49	8.40	8.30	8.70	8.62	6.25
5	SLC39A1	Solute carrier family 39 member 1 (Zinc transporter ZIP1)	169.84	96.51	63.39	68.11	90.07	62.21	83.81
6	SLC30A9	Solute carrier family 30 member 9 (Zinc transporter 9)	1.37	1.12	1.24	1.58	1.24	1.10	1.10
7	SLC30A7	Solute carrier family 30 member 7 (Zinc transporter 7)	1.96	1.11	1.14	1.67	1.46	2.03	1.44
8	SLC30A6	Solute carrier family 30 member 6 (Zinc transporter 6)	0.89	1.05	1.06	0.94	1.11	1.38	0.93
9	SLC30A5	Solute carrier family 30 member 5 (Zinc transporter 5)	0.85	1.20	1.20	0.94	1.31	1.30	1.36
10	SLC30A2	Solute carrier family 30 member 2 (Zinc transporter 2)	14.44	12.71	10.37	12.75	14.66	13.42	9.99
11	SLC30A1	Solute carrier family 30 member 1 (Zinc transporter 1)	2.88	2.39	1.52	1.82	2.47	1.63	1.70
12	SLC26A6	Solute carrier family 26 member 6 (chloride-bicarbonate exchanger)	2.50	2.82	2.77	2.97	4.08	4.83	2.83
13	SLC26A11	Solute carrier family 26 member 11 (chloride-bicarbonate exchanger)	1.89	2.31	2.34	2.35	3.34	2.39	1.55
14	SLC22A8	Solute carrier family 22 member 8 (Organic cation transporter)	9.04	10.59	12.87	8.80	11.79	4.89	11.21
15	SLC22A6	Solute carrier family 22 member 6 (Organic cation transporter)	22.11	24.18	14.19	15.98	17.75	19.86	14.23
16	SLC22A5	Solute carrier family 22 member 5 (Organic cation transporter)	0.79	0.86	0.76	0.63	0.68	0.54	0.47
17	SLC22A3	Solute carrier family 22 member 3 (Organic cation transporter)	8.86	11.13	8.70	6.63	9.70	12.58	10.05
18	SLC22A15B	Solute carrier family 22 member 15B (Organic ion transporter)	11.15	11.08	9.40	11.31	11.34	10.87	10.04
19	SLC13A5	Solute carrier family 13 member 5 (Na^+^-dependent carboxylate and sulfate transporter)	15.67	12.67	9.84	13.95	11.28	34.18	15.09
20	SLC12A9	Solute carrier family 12 member 9 (Cation-coupled Cl^−^ cotransporter)	10.11	9.59	10.29	9.13	11.05	8.53	6.02
21	SLC12A6	Solute carrier family 12 member 6 (Cation-coupled Cl^−^ cotransporter)	5.00	4.57	8.13	5.15	7.42	5.91	4.65
22	SLC12A2	Solute carrier family 12 member 2 (Na^+^/K^+^/2Cl^−^ cotransporter)	13.86	25.66	17.36	11.79	17.32	74.29	26.06
23	SLC10A6	Solute carrier family 10 member 6 (Ileal sodium/bile acid cotransporter)	56.69	37.14	40.56	46.31	43.40	16.17	27.24
24	SLC10A3	Solute carrier family 10 member 3 (Ileal sodium/bile acid cotransporter)	3.08	2.52	3.42	1.81	2.65	2.11	1.50
25	SLC10A2	Solute carrier family 10 member 2 (Sodium/bile acid cotransporter)	51.12	89.37	36.31	50.32	54.28	34.86	36.86
26	SLC9A8	Solute carrier family 9 member 8 (Sodium/hydrogen exchanger 8)	1.12	1.71	1.74	1.75	1.84	3.16	1.95
27	SLC9A9	Solute carrier family 9 member 9 (Sodium/hydrogen exchanger 9)	2.64	2.88	3.02	2.51	3.28	4.20	2.28
28	SLC9A3R1	Solute carrier family 9 member 3 regulator 1 (Na^+^/H^+^exchange regulatory cofactor NHE-RF1)	29.13	40.50	32.40	28.00	37.29	58.53	32.90
29	SLC9A2	Solute carrier family 9 member 2 (Na^+^/H^+^ exchanger)	0.13	0.34	0.27	0.16	0.54	5.04	0.56
30	SLC8A2	Solute carrier family 8 member 2 (Sodium-calcium exchanger)	0.68	1.60	1.72	1.22	2.14	4.33	1.48
31	SLC8A1	Solute carrier family 8 member 1 (Sodium-calcium exchanger)	0.31	0.63	0.38	0.62	0.46	0.50	0.46
32	SLC5A8	Solute carrier family 5 member 8 (Sodium-dependent vitamin transporter)	2.32	2.59	2.65	2.35	2.78	3.90	1.77
33	SLC5A3	Solute carrier family 5 member 3 (Sodium/myo-inositol cotransporter)	5.42	10.56	8.26	7.93	11.63	12.35	8.77
34	SLC5A1	Solute carrier family 5 member 1 (Sodium/glucose cotransporter)	23.56	227.19	181.73	82.43	160.33	283.00	110.62
35	SLC4A1	Solute carrier family 4 member 1 (Sodium bicarbonate cotransporter)	1.68	1.79	1.72	1.50	1.84	1.56	1.00
36	SLC4A10	Solute carrier family 4 member 10 (Sodium bicarbonate cotransporter)	2.04	2.40	2.23	1.81	2.60	1.29	1.37

## Data Availability

Data are contained within the article or Appendix A.

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
