# Peer review of "Effects of Different Salinity Stress on the Transcriptomic Responses of Freshwater Crayfish (Procambarus clarkii, Girard, 1852)"

_biology, 2024, doi:10.3390/biology13070530_

Round 1
Reviewer 1 Report
Comments and Suggestions for Authors
the suggestions are in the document
-in line 149 change P. clarkii to cursive
-homogenize in the text "up-regulated" or "upregulated"

The document is written in understandable English, but requires attention to certain details that are indicated in the document.
Reviewer 2 Report
Comments and Suggestions for Authors
Authors analyzed the Procambarus clarkii, a valuable freshwater aquaculture species, response on the environmental salinity stress. They developed transcriptomic data resources and identify the DEGs associated with various metabolic pathways and their expression pattern. The research is valuable. However, it still need a minor revision.
1. What the resource of the crayfish? How to determine the taxonomy of the species?
2. Why chose the salinity concentration of 6 and 18ppt? Authors should clarify the purpose of the design.
3. Authors should pay attention to the quality of the figures, ensure the quality of the figures is 300DPI at less or above, and clear when expanded.
4. The title of 4.4.2. and 4.4.3 are same. The author should change one of them.
5. The ion symbol should be changed to superscript in Discussion part, such as “HCO3-”.
6. Authors should note that the Latin names of crayfish throughout the text need to be italicized
7. Authors can clarify what are importance of the study and make the research more meaningful.
Comments on the Quality of English Language
The english is well written, but still need some proofread.
Reviewer 3 Report
Comments and Suggestions for Authors
Procambarus clarkii, commonly known as the red swamp crayfish, is a prevalent freshwater crustacean in China with a vast market and high economic value. In this study, the authors aim to identify genes and pathways associated with salinity tolerance in crayfish by comparing transcriptomes at different time points under various salinity stress conditions. The goal is to provide scientific assistance for addressing the challenges of crayfish farming in saline-alkali soils.
Although the research direction of this paper holds practical significance, the writing quality needs further improvement, and the language requires refinement. Additionally, the study is relatively fundamental, primarily focusing on transcriptome sequencing and analysis. This approach is common and tends to be somewhat monotonous, with limited innovation. The validation methods are merely repetitive and mechanical, lacking deeper investigation. For instance, it would be beneficial to re-expose a new batch of crayfish to salinity stress and verify the expression levels of relevant genes. Furthermore, the authors could consider verifying gene expression from other aspects, such as protein expression. Establishing a salt-tolerant crayfish population and conducting expression verification of relevant genes in both salt-tolerant and non-tolerant populations would also be valuable.
Here are some specific issues and suggestions:
Introduction
1. Line56: Replace "has led to an increase in sea and estuarine levels" to "has led to rising sea and estuarine levels".
2. Line60: Delete "However", as there is no contrasting relationship.
3. Line63: Remove the redundant phrase "worldwide" as "global problem" already conveys this meaning.
4. Line67: Replace "an important environmental factor directly affecting the metabolism, " to "an important environmental factor directly affecting metabolism, ".
5. Line73: Replace "through increased energy metabolism " to "through heightened energy metabolism."
Materials and Methods
1. Exposure and sample collection: The text does not describe the sampled tissues, and different tissues significantly affect experimental results. Has the author considered the main organs of crayfish in response to salinity stress?
2. Exposure and sample collection: Was feed provided during the experiment? How can the potential impacts of starvation or water quality changes be excluded in the 72-hour stress experiment without feeding or water change? Has the author considered setting up control groups for each time point? Please provide descriptions of both the experimental and control groups.
Results
1. Quantitative real-time PCR: The trends of the five genes SOD-4, CHER, DUOX, RGN, TPI1B in Figure 7 do not align with the transcriptome results. Please provide a reasoned discussion and explanation.
2. Figure 7: Why is there no significant analysis included for the comparison of relative expression levels?
3. Figure 7: Some significant differences between groups in the figure could potentially be represented more clearly with breakpoints, which might better illustrate the trends.
Comments on the Quality of English LanguageThere is still room for improvement in the wording of this article. Please pay attention to the consistency of tenses before and after citations, as well as the length and redundancy of sentences and phrases. The professional term "P. clarkii" is used too frequently; consider using "crayfish" instead. Some sentences are overly colloquial; try utilizing the passive voice to enhance sentence structure.
